# Organizational Cynicism and Its Impact on Organizational Pride in Industrial Organizations

**DOI:** 10.3390/ijerph16071203

**Published:** 2019-04-03

**Authors:** Omar Durrah, Monica Chaudhary, Moaz Gharib

**Affiliations:** 1Management Department, College of Commerce and Business Administration, Dhofar University, Salalah 221, Oman; mnagib@du.edu.om; 2Department of Humanities and Social Sciences, Jaypee Institute of Information Technology, Noida, Uttar Pradesh 201309, India; monicarana@gmail.com

**Keywords:** organizational cynicism, cognitive cynicism, affective cynicism, behavioral cynicism, organizational pride, emotional pride, attitudinal pride, industrial companies, Oman

## Abstract

Organizational cynicism has been a topic of discussion and debate among employees and top management. The purpose of this study is to find out the relationship between organizational cynicism and organizational pride. Precisely, the objectives are to identify and measure organizational cynicism among employees in industrial organizations; to determine and measure the degree of organizational pride among employees in industrial organizations and to study the effect of organizational cynicism on the organizational pride of employees in industrial organizations. In this empirical research, the study population was employees of industrial organizations of Oman. Using a purposive sampling technique, nine industrial organizations from Oman were picked. With the help of structured questionnaire, data from 350 respondents was obtained. Structural equation modeling was used through Amos version 25.0 for data analysis. The results reveal that the two dimensions of organizational cynicism (affective cynicism and behavioral cynicism) have a significant and negative impact on emotional pride, while cognitive cynicism does not significantly effect emotional pride. The study results indicate that the one dimension of organizational cynicism (affective cynicism) has a significant impact on attitudinal pride, while the rest of the other dimensions (cognitive cynicism, behavioral cynicism) do not have a significant effect on attitudinal pride. The limitations and implications of the research are also discussed.

## 1. Introduction

Organizational cynicism is an old phenomenon, despite the modernity of dealing with it by researchers and practitioners, organizational cynicism is a feeling of dissatisfaction towards the organization, and employees believe that the organization’s management lacks honesty, justice, and transparency [1]. Dean and colleagues defined organizational cynicism as a negative attitude (negative attitude of aggravation) towards the organization [2]. There are other studies which explored other dimensions of organizational cynicism like an attitude of unfriendliness, lack of honesty by organization, disturbance, dissatisfaction, and hopelessness about the organization [3]. Organizational cynicism is widespread among organizations globally; there have been studies in USA, Europe, and Asia [2,4,5,6].

As academicians, we are always in search of studies which can be helpful for people to understand, analyze, and perform better. There is always a need to study different aspects of management in different cultural environments [7]. In his famous cultural study, Hofstede drew the majority of data from developed economies like that of USA, UK, and Europe [7]. Other than this developed world, a large portion of the world also belongs to developing, less developed, and emerging economies. Oman, as one of the GCC countries (Gulf Cooperation Council), is an emerging economy, which is growing at a very fast rate. Between 2006 and 2013, Oman’s economy had grown by 71%, from $47 billion in 2006 to $80.5 billion in 2013 [8].

Also, in this digital age, employee health has been understood by researchers as an important element for organizational growth. New digital trends such as cloud computing, mobile web services, and social media are radically changing work place dynamics [9]. Mental health is described by the World Health Organization (WHO) as, “an illness is a serious public health problem around the world, with about 450 million people having a mental health disorder” [10]. ‘Digital Depression’ is the term given to the feeling of being overwhelmed and overworked by technology [11]. New evolving technologies have come with a cost; the pressure to be constantly available via technologies constitutes a major source of stress, increasing the risk of experiencing prolonged work stress and its adverse consequences on employee health and well-being, such as a burnout [12]. There is a need to study employee health in this new digital paradigm.

It is evident from a vast review of literature that there was a lack of context specific to different dimensions of organizational cynicism, especially in an emerging economy like Oman. Based on this, the main purpose of this study is to investigate the relationship between organizational cynicism and organizational pride in Oman. The following three objectives were formulated:To identify and measure organizational cynicism among employees at industrial organizations in Oman.To determine and measure the degree of organizational pride among employees at industrial organizations in Oman.To study the effect of organizational cynicism on the organizational pride of employees at industrial organizations in Oman.

In this empirical study, the effect of organizational cynicism on organizational pride in the organizations in Oman is studied in detail. The researchers address the crucial aspects of organizational cynicism constructs and how these constructs connect with organizational pride. The paper is organized as a systematic study. First, in the literature review section, the origin of ‘cynicism’ and ‘organizational cynicism’ is discussed. Then, a model of cynicism in organizations is proposed and a number of related issues are presented, then organizational pride is explained and its dimensions. In the next section of research methodology, a detailed explanation of sampling and data collection process is presented. The paper then moves to the Data Analysis section and finally the Conclusion, along with implications, is presented. The paper ends with highlighting a few of the study limitations.

## 2. Literature Review

### 2.1. Originality

Past literature revealed that there was a lack of context specific research regarding different dimensions of organizational cynicism, especially in an emerging economy like Oman. Oman, as one of the GCC (Gulf Cooperation Council) countries, is an emerging economy which is growing at a very fast rate. This study is the first one to explore the effect of organizational cynicism on organizational pride of employees.

### 2.2. Origin of Cynicism 

The word cynicism can be traced back to fourth century Greece. A group of philosophers who called themselves as followers of Antisthenes very openly questioned the existence of government and religious institutions [2]. Many believed that these followers Antisthenes flouted popular opinion or public convictions simply for the sake of doing so, and deemed these followers as disciples of the dog, or Cynics [13]. However, as time progresses (third century), Cynicism was revived as a school of thought and propagated the idea of mockery of convention and tradition and prevailing beliefs and modes of behavior [14]. 

### 2.3. Organizational Cynicism

With time, organizational cynicism emerged as the new paradigm of employer–employee relations [15]. Researchers found that a significant percentage of employees were highly cynical about their organizations [4,16]. In the simplest words, organizational cynicism arises when employees lack confidence in their organization and feel that the organization cannot be trusted. Most definitions of organizational cynicism are associated with emotions such as disillusionment and anger [17]. 

Cynicism has been studied extensively from a psychological perspective. From an organizational point of view, Niederhoffer in his 1967 study was the first to analyze and measure cynicism in police officers [18]. Another set of researchers deduced that cynicism might affect organizations and their members through the “break down [of] authority” [19]. However, it was during the 1990s when both practitioners and academicians started paying some attention to cynicism within organizations. During this time, organizational cynicism studies were considered in the first stage of scientific research [20]. There were many studies which focused on the systematic examination of organizational cynicism as a construct that directly influences attitudes, beliefs, and behaviors [2,4,21,22,23].

### 2.4. Organizational Cynicism Dimensions

Researchers have begun to recognize the important effect that cynicism can have on organizations [24]. Organizational cynicism is seen as an attitude or belief [25]. The studies conducted through the 1980s and 1990s generally studied cynicism under three aspects:-Discussed as a personality trait [26].-In the light of industry-level environmental causes of cynicism like sacking employees or cutbacks [22]. -Causes that are under the direct control of individual organizations [4,22], where the organizational cynicism includes a stable personal component as well as the situational component [2].

Then, during the 2000s, more in-depth studies contributed to different aspects of organizational cynicism. In a very exhaustive study by James, organizational cynicism is divided into five distinct structures, namely [24]:-“Personal cynicism” is something which is a personality trait of an individual. -“Societal cynicism” is having negative feelings towards society in general. -“Cynicism towards change” is about an employee’s specific distrust towards any organizational change [20] and the nature of a career [15]. -“Work cynicism” is explored as a component of burnout [27]. Work cynicism means holding a secluded and indifferent attitude to one’s work and the predisposition to assess one’s own performance at work in negative terms [28,29]. The studies also found that some form of work-related cynicism might act as a coping strategy for employees [21].-“Employee cynicism” refers to behavioural outcomes and negative attitudes of employees [30]. Just like other dimensions of organizational cynicism, employee cynicism also has an opposite effect on productivity and organizational processes [31]. For example; an employee may involve himself in loafing rather than doing his work [32]. This is a first step toward the more extreme stage of work withdrawal [33].

Organizational cynicism comprises of three distinct dimensions, namely [2,34,35,36]:-‘Cognitive cynicism’ refers to lack of sincerity, honesty, and justice in the organization, where cognitive cynicism is accessible when staff feel that their corporation does not esteem their endeavours or care about every one of them, and therefore may be unlikely to make their best efforts for their corporation [35]. Workers facing cognitive cynicism think that principles are often sacrificed for expedience, and that duality, guile, and personal interest are common in their firms [35]. Bernerth and colleagues found that employees’ perceptions of cognitive cynicism are negatively associated with organizational commitment [37]. Similarly, Abraham indicated that cognitive cynicism reduces the performance in the organization [21].-‘Affective cynicism’ refers to emotional and sentimental responses towards the organization, and involves psychological reactions such as aggravation, tension, anxiety, and discomfort; where the cynics feel disrespect and frustration towards their firms [38]. Mishra and Spreitzer indicated that actual cynics experience different emotions such as moral outrage, anger, and hatred towards their employing organization [39]. Affective cynicism is accompanied by the arrogance as the cynical employees believe that they have the superior understanding and outstanding knowledge of the things [39].-‘Behavioral cynicism’ refers to critical expressions and negative attitudes frequently used in the organization. Behavioral cynicism consists of sarcastic humour, criticism of the organization, unfavorable non-verbal behavior, negative interpretations of attitudes in the organization, and cynical predictions about the organization’s action in the future [35]. The behavior of cynical employees includes humorous and stinging attitudes and bad mouthing towards their organization, in addition, employees who ridicule their organization and senior management tend to be less likely to make efforts for their jobs [40]. These employees exhibit poor work performance in the organization [41].

Organizational cynicism is reported to have a negative impact on employee performance. In a 2008 study in the USA, responses from 1256 full-time employees and 2143 full-time state employees from a variety of industries were taken. This study concluded that a cynical employee’s performance was highest when perceived support was at moderate levels only. Conversely, performance for cynics was lowest when perceived support was either high or low [42].

### 2.5. Organizational Pride 

Organizational pride is essentially a psychological structure examined in psychology studies with a particular focus on the relationship of employees with their organizations [43]. Mischkind defined organizational pride as a positive feeling by the employee in his institution [44]. Organizational pride consists of feelings of admiration, importance, and value based on evaluations of status made by staff [45,46].

The concept of organizational pride has attracted the interest of both practitioners and management scientists because of its importance as a driver of positive job behaviors and the main differentiator in competition [47], as strategic assets of the company [40], and the vital factor for success of the business [48,49].

Organizational pride is referred to as an encouraging constructive work environment that needs high social recognition with the organization [50]. Gouthier and Rhein put forward that organizational pride studies need more scientific attention and it is expected to be a vital factor for the success of a business [49]. The more positively workers evaluate their organization, the more they feel committed to it and hence experience organizational pride [51].

### 2.6. Organizational Pride Dimensions

Contrary to what organizational cynicism usually means, organizational pride has positive connotations. Similarly to organizational cynicism, organizational pride is also considered an emotion [52]. Organizational pride is found to have an affirmative and significant relationship with job satisfaction [53]. The very first dimension in which organizational pride has been studied by scholars is related to self-respect and self-worth of person. Organizational pride is psychological traits that can be tactfully used to enhance employees’ motivation. Organizational pride is studied based on construction of the membership of the staff group [54].

Employees might develop a constant interior-pride attitude toward their organizations [55]. Increased organizational pride is believed to negatively affect turnover [49,50]. Previous studies suggest that organizational pride increases resistance to stress and reduced intentions of turnover. Not just reducing the negative emotions, organizational pride positively affects the decision to stay with a company and encourages staff commitment [48], autonomy, team support, and considerations of the supervisor among subordinates [50].

According to a study by Germany’s Federal Ministry of Education and Research, there are two types of organizational pride [49]. The first type of organizational pride depends on the perception of a successful event related to the firm. Under this type of organizational pride, employees feel short, continuous affective emotions of pride. The other type of pride depends on the general perception of the firm. Under this type of organizational pride, employees have a moral and permanent attitude of pride.

Kraemer and Gouthier divided organizational pride into two types, the first is emotional pride and the second is attitudinal pride [50]. Emotional pride is pride which is strong but discrete. It is also described as a short-lived mental experience. Attitudinal organizational pride on the other hand is durable and can be learned. In contrast to emotional pride, organizational pride in attitudes is collective, resulting from the staff desiring to belong to the company [56].

### 2.7. Need of the Study

Most of the definitions we studied relate cynicism with negative emotions. Certainly, such definitions make it difficult to place organizational cynicism on a common ground with organizational pride. However, there is a need to understand the elements of organizational cynicism in more depth as concluded by Brett Waring, who described “organizational cynicism as an attitude consisting of the futility of change along with negative attributions of change facilitators” [57]. With this thought, we can move forward and can relate organizational cynicism with organizational change. In a very helpful study by [57] in which studied elements of cynicism in the US Army were studied, it was found that “cynicism does not exist in an attitudinal vacuum, but resides on a sliding scale with pride, skepticism, sarcasm, and pessimism. These negative thoughts and feelings lead to misapplied, misplaced, or even denied pride [57]. Therefore, cynicism fills a gap left where pride either should have flourished or diminished as expectations were not reconciled with reality.” An employee with this denied pride will develop an emotional gap that creates a chance to promote a negative stance as an employee drifts away from positive thinking.

Despite many multi-disciplinary studies, we cannot ignore the fact that organizational cynicism must be impacting an employee’s emotional and attitudinal pride. With extensive literature review, we can say that there is no single study which explores the relationship between organizational cynicism and organizational pride of employees. This original study contributes to the extant literature in the area and provides very insightful analysis of organizational cynicism’s effect on the organizational pride of employees at industrial organizations in Oman.

### 2.8. Hypotheses of Research 

According to the previous comprehensive review of literature and objectives of research, the following hypotheses are formed to analyze the proposed impact of organizational cynicism on organizational pride:

H_1_: Cognitive cynicism has a significant impact on emotional pride.

H_2_: Affective cynicism has a significant impact on emotional pride.

H_3_: Behavioral cynicism has a significant impact on emotional pride.

H_4_: Cognitive cynicism has a significant impact on attitudinal pride.

H_5_: Affective cynicism has a significant impact on attitudinal pride.

H_6_: Behavioral cynicism has a significant impact on attitudinal pride.

### 2.9. Research Methodology

The present study aims to determine the effect of organizational cynicism on organizational pride. The study population consisted of administrative staff from nine industrial organizations in Oman as shown in Table 1.

Out of 420 distributed questionnaires distributed according to purposive sampling technique, 350 questionnaires were completely filled by respondents, so they were valid for statistical analysis. To collect data for the present study, the survey method was employed using paper questionnaires which were distributed to the respondents during official working hours in the examined organizations. Research literature was reviewed in order to develop the questionnaire, which consisted of three sections. The first section includes demographic data of respondents, while the second section includes the questions about organizational cynicism based on the scale of Dean and colleagues [2], which used by (Erarslan, et al., 2018; Nafei and Kaifi, 2013;) [36,58] in their studies. Scale of organizational cynicism is comprised of the three dimensions of cognitive cynicism, affective cynicism, and behavioral cynicism. It consists of 12 statements, wherein every dimension has four items (see Table A1). The third section includes questions about the organizational pride was measured by seven items developed according to scale of Gouthier and Rhein, (2011) [49], which used by (Swanson and Kent, 2017 and Welander, et al., 2017) [59,60] in their studies. The organizational pride scale includes two dimensions of emotional pride (four items) and attitudinal pride (three items) (see Table A1). 

A five-point Likert-type scale was used in building the survey question format, the responses ranked from 1 (strongly disagree) to 5 (strongly agree). The data obtained from the survey were analyzed using the Amos version 25.0 software. Most statistical tests like descriptive statistics, reliability, correlation, and exploratory factor analysis (EFA) were analyzed by SPSS program, while confirmatory factor analysis (CFA) and structural equation modeling (SEM) were used through an AMOS program to test the hypotheses.

## 3. Data Analysis

To analyze the data of this study, SPSS (IBM Corp., Armonk, NY, USA) was used for a set of statistical methods such as descriptive statistics, reliability, correlation between variables, multicollinearity of independent variables, and exploratory factor analysis (EFA) to determine the dimensions of the study variables. [61]. AMOS was used to test the hypotheses of the study, confirmatory factor analysis (CFA) was applied, where it is a multivariate statistical procedure that is used to test how well the measured variables represent the number of constructs [62]. CFA was used to confirm the exploratory factor model by determining the goodness of fit between hypothesized model and sample data [63]. Structural equation modeling (SEM) was applied on the data to test of study hypotheses, it is a methodology for representing, estimating, and testing a number of relationships between variables [64]. The SEM approach is used to validate the research model, SEM is a largely confirmatory, rather than exploratory, technique [65]. The results of the quantitative survey are tabulated and discussed in this section. Table 2 shows the details of the sampled respondents’ demographics.

In the current study, frequency and percentage techniques were used for the demographic variables of the participants as shown in Table 2. Where 81.1% of respondents were male and 18.9% of respondents were female. Among those sampled, 76% were married, and 60.57% had a diploma or less. Most of the respondents’ were aged between 30 to 50 years, also most of the respondents’ experience lies between 5 to 10 years. 

This table shows the prevalence of organizational cynicism among respondents and their awareness of organizational pride towards their companies. Where there was a convergence between males and females in terms of organizational cynicism, while males showed more pride than females. Regarding marital status, it was found that single people had a higher rate of cynicism than married people, while conversely, married people were more prideful than single categories towards their organizations. Regarding age groups, the first age group was the most prevalent category for organizational cynicism among workers. While the last age group showed more pride. In terms of qualifications, diploma holders were the most cynical, while postgraduate holders were the most prideful of their organizations. Finally, those with 5 to 10 years of experience showed a high level of cynicism towards their companies, while those with more than 10 years’ experience showed pride in their organizations.

Means and standard deviations were used to determine the levels of organizational cynicism and organizational pride. Analysis of Pearson correlation was used to investigate the relationships between study variables. The coefficient of Cronbach’s has been used to estimate the scales reliability. The mean, standard deviation, skewness, kurtosis, correlation, and scales reliability are provided in Table 3, where this table shows the dimension of cognitive cynicism was the highest mean (2.71) of organizational cynicism dimensions, followed by affective cynicism (2.45), and the lowest mean was behavioral cynicism (2.20). Regarding dimensions of organizational pride, attitudinal pride was the highest mean (3.90), while emotional pride was the lowest mean (2.74). Standard deviation values of all the study variables were less than (1) except quiet emotional pride. Values of Cronbach’s Alpha range between (0.720 and 0.875), these values are within the acceptable range (>0.60). The relationships between most of the study variables were significant and correlated at a level of 0.01. Values of skewness and kurtosis exist in the acceptable range {−3 to, +3} [66].

Table 4 shows the prevalence of organizational cynicism in surveyed companies, it was found that this phenomenon is widespread low-grade (2.46); this is an indication of the availability of healthy working environment in these companies in general. Octal is the most prevalent company in which the phenomenon of organizational cynicism prevails (2.92). In contrast, the company Dhofar Cattle Feed is the least (1.93), and this may be due to the nature and type of work in each company and the extent of the relationship between management and employees. The same table shows the prevalence of organizational pride among workers in these companies, it has been shown that this phenomenon is available to a medium degree among employees in these companies (3.32). The employees of Salalah Methanol company show the most pride (3.83), the reason is that this company is one of the best companies in the Sultanate of Oman with a good reputation and it provides great services and benefits to its employees in addition to high salaries, while employees at Octal company showed the least pride (2.71), this is due to the workers’ dissatisfaction with their job, the difficulty of the tasks entrusted to them, and the absence of incentives and rewards.

The condition of multicollinearity between independent variables has been achieved because all the values of tolerance > 0.05 and all values of variance inflation factors < 10 as shown in Table 5 [67].

To determine the organizational cynicism dimensions and organizational pride dimensions we used the exploratory factor analysis (EFA) through the principal components by the Varimax method as shown in the Table 6. Exploratory factor analysis (EFA) revealed the existence of five variables through the entered items as shown in Table 4. Three dimensions of them for organizational cynicism (the first is cognitive cynicism, the second is affective cynicism, and the third is behavioral cynicism) and two of them for organizational pride (i.e., emotional pride and attitudinal pride). 

In view of Table 6, it is noted that all conditions of (EFA) have been achieved. (KMO = 0.892 > 0.60, Bartlett’s Test = 5331.044, Sig. = 0.000 < 0.05, Cumulative Variance = 70.69 > 60). Eigen values for every factor was greater than one.

Exploratory factor analysis (EFA) revealed five constructs; this gave way to confirmatory factor analysis (CFA). To conduct this analysis was used AMOS program to confirm the exploratory factor model by determining the goodness of fit between hypothesized model and sample data as shown in Figure 1.

Structural equation modeling (SEM) through AMOS was conducted to show the effect the dimensions of organizational cynicism on the dimensions of organizational pride. In Figure 2, sub-scales of organizational cynicism (cognitive cynicism, affective cynicism, and behavioral cynicism) were added as independent variables, whereas two dimensions of organizational pride (Emotional pride, and attitudinal pride) were included as dependent variables. 

Table 7 shows fit indices of the final model of the current study. According to Figure 1 and Figure 2 and Table 6, the results mentioned indicate a good fit of the model that was tested according to fit indices and criteria, where all the results were within the acceptable values [68,69].

Table 8 shows the results of hypothesis testing for all dimensions of organizational cynicism (i.e., Cog, Aff, and Beh) with organizational pride (i.e., Emo and Att). We can see that organizational cynicism dimensions (Aff and Beh) have a significant impact on emotional pride (Emo) and the third dimension (Cog) has no significant impact on emotional pride (Emo). The results also show that one dimension of organizational cynicism (Aff) has a significant impact on attitudinal pride (Att) and the other dimensions (Cog and Beh) have no significant impact on attitudinal pride (Att). Thus, these three hypotheses were accepted (H_2_, H_3_, and H_5_), while the other hypotheses were rejected (H_1_, H_4_, and H_6_).

## 4. Discussion 

As stated earlier, objective of this study is to investigate the impact of organizational cynicism dimensions (cognitive cynicism, affective cynicism, and behavioral cynicism) on the organizational pride dimensions (emotional pride and attitudinal pride) of employees in the industrial organizations in Oman. The literature reflects several studies on the multiple aspects, but the researchers in this paper did not find any study directly covered the relation between organizational cynicism and organizational pride. This study contributes to the limited literature available in this area. So far, very few studies were conducted at this detailed level. Moreover, the gulf region is still unexplored from the employees’ point of view in organizations.

The present study provides thorough insights about organizational cynicism among the employees in industrial organizations in Oman and its relevance to organizational pride despite the variety of problems related to them. This study is important for researchers who want to understand the organizational cynicism dimensions and their impact on organizational pride.

In general, organizational cynicism has a negative relationship with many variables such as employee’s performance [35], work performance in the organization [41], organizational commitment [58], quality of life [70], and change [71]. This negative relationship was confirmed in this study, which means increased organizational cynicism reduces organizational pride.

Both affective and behavioral dimensions were found to have a significant and negative impact on emotional pride; whereas the results revealed that cognitive dimension had no influence on emotional pride. This negative impact of affective and behavioral cynicism on emotional pride reveals the relationship between the variables and also reflects the feeling of employees’ tension and anger toward organizations that are not fulfilling their promises and betraying them in different ways. This breach of contract becomes the reason for organizational cynicism among employees, badly affecting their emotional pride. While the negative influence of affective cynicism on attitudinal pride is due to staff with high negative effects tending to be more sensitive to events or attitudes that support their beliefs and values. The self-esteem of the individual also affects the growth of his feelings of frustration, the events in the organization are viewed pessimistically, which reduces the extent of his association and pride in his organization.

## 5. Implications

The current study has some vital implications that indicate its importance, where there is a lack of a comprehensive model of organizational cynicism in literature; this study attempts to improve organizational behavior in a more holistic manner. This study helps policymakers of organizations to encourage employees to show their feelings and emotions during work and to not suppressing them, reducing the negative effects of organizational congestion, such as indifference, control of frustration at work, decisions to resign, loss of trust in others, and tension of personal relationships within the organization. Organizations must strive to achieve good relations with employees in a spirit of passion and mutual trust and not resort to the application of punitive policies with workers who show anger and boredom with administrative and regulatory policies.

The present study contributes to development of organizational pride through the preparation of a record that includes the excellent successes of the company. The continuous awareness of the employees, which is achieved through the holding of meetings and seminars aimed at raising employee awareness and clarifying achievements and outstanding successes of the factory and services provided to the community and other factors that can contribute to generating feelings of organizational pride among employees.

## 6. Limitations and Future Research 

Although the present study is a pioneering study in its contextual subject area, this research cannot be completely generalized as it is lined up with some limitations. Firstly, the research is limited to industrial organizations only, hence it cannot be generalized to all the organizations in Oman. Secondly, it was quite difficult to collect the data of a larger sample frame because of employee schedules and the availability of free time, which restricted researchers to a limited sample size in this study.

On the other hand, it would be useful to examine the relation the organizational cynicism with the organizational pride at huge companies. A comparative study can be made between public and private companies. This topic can be applied in different areas other than the industrial sector (i.e., health, security, hospitality, educational institutions, etc.).

## 7. Conclusions

The current study has explored the impact of three dimensions of organizational cynicism on organizational pride in the Omani context with reference to the industrial organizations. The results offer an understanding of organizational behavior in industrial organizations in the Sultanate of Oman. The present study provided managers an insight to understand how to reduce different dimensions of organizational cynicism and increase the organizational pride of staff. Therefore, it is advisable according to the current study that the concepts of organizational cynicism and pride are without any doubt essential factors for the industrial sector particularly, and for other sectors generally.

## Figures and Tables

**Figure 1 ijerph-16-01203-f001:**
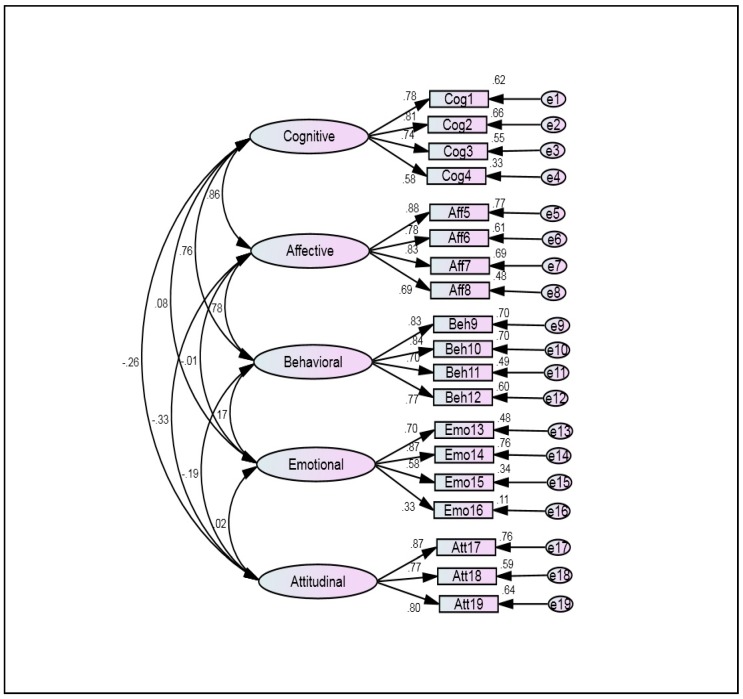
Exploratory factor analysis of measurement model.

**Figure 2 ijerph-16-01203-f002:**
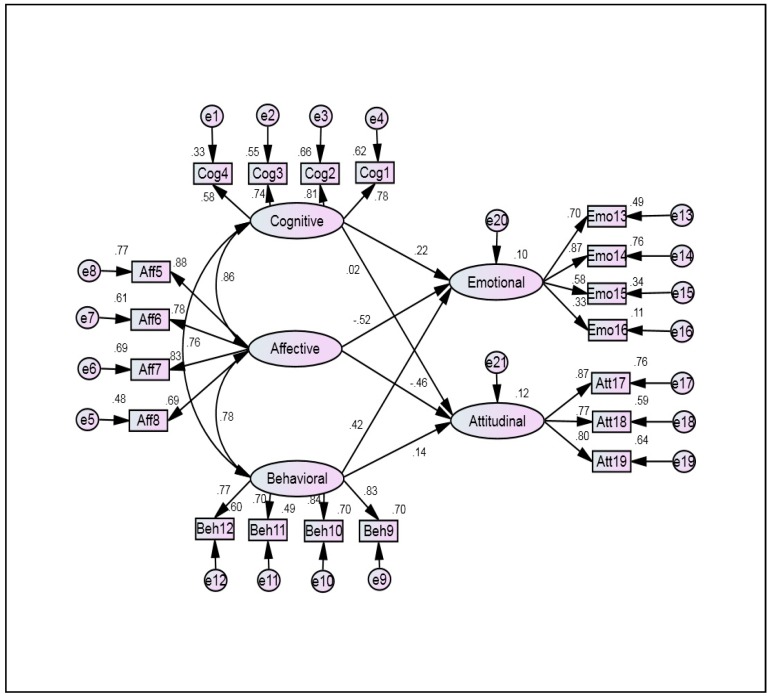
Structural equation model.

**Table 1 ijerph-16-01203-t001:** Organizations in the study.

No.	Company Name	No. of Participants	Percentage
1	Salalah Methanol Company	97	27.71
2	Port of Salalah	62	17.72
3	Salalah Mills Co.	53	15.14
4	Raysut Cement Company	40	11.43
5	Dhofar Cattle Feed	27	07.71
6	Octal	24	06.86
7	Dhofar for Power	20	05.71
8	Oman Oil Company	15	04.29
9	Oman National Factory for Printing and Packaging	12	03.43
	Total	350	100.0

**Table 2 ijerph-16-01203-t002:** Demographic data characteristics.

Variables	Categories	Frequency	Percentage	Organizational Cynicism	Organizational Pride
Sex	Male	284	81.10	2.45	3.47
	Female	66	18.90	2.47	3.18
Marital Status	Single	84	24.00	2.51	3.14
	Married	266	76.00	2.42	3.51
Age	Less than 30 years	79	22.57	2.56	3.11
	Between 30 and 50	254	72.57	2.44	3.37
	More than 50	17	04.86	2.38	3.48
Qualification	Diploma and less	212	60.57	2.52	3.28
	Bachelor	119	34.00	2.42	3.25
	Postgraduate	19	05.43	2.44	3.43
Experience	Less than 5 years	59	16.86	2.41	3.16
	Between 5 and 10	162	46.29	2.63	3.33
	More than 10	129	36.85	2.34	3.47
Total		350	100%	2.46	3.32

**Table 3 ijerph-16-01203-t003:** Descriptive statistics, correlations, and reliability.

Variables	Mean	SD	Kurtosis	Skewness	Cognitive	Affective	Behavioral	Emotional	Attitudinal
Cognitive	2.71	0.986	−0.863	0.094	(0.819)				
Affective	2.45	0.913	−0.100	0.693	0.695 **	(0.875)			
Behavioral	2.20	0.901	0.080	0.915	0.660 **	0.678 **	(0.866)		
Emotional	2.74	1.14	−1.15	0.121	0.075	−0.072	0.113 *	(0.720)	
Attitudinal	3.90	0.849	1.98	−1.24	−0.219 **	−0.278 **	−0.166 **	0.071	(0.852)

Notes: *n* = 350, * *p* < 0.05, ** *p* < 0.01, SD: standard deviation, (): reliability.

**Table 4 ijerph-16-01203-t004:** Prevalence of organizational cynicism and organizational pride in companies.

No.	Company Name	Organizational Cynicism	Organizational Pride
1	Shalala Methanol Company	2.49	3.83
2	Port of Salalah	2.03	3.32
3	Salalah Mills Co.	2.70	3.29
4	Raysut Cement Company	2.77	3.51
5	Dhofar Cattle Feed	1.93	3.34
6	Octal	2.92	2.71
7	Dhofar for Power	2.61	3.22
8	Oman Oil Company	1.97	3.44
9	Oman National Factory for Printing and Packaging	2.80	3.28
	Total	2.46	3.32

**Table 5 ijerph-16-01203-t005:** Multicollinearity test of independent variables.

Independent Variables	Tolerance > 0.05	Variance Inflation Factor VIF < 10
Cognitive	0.451	2.216
Affective	0.432	2.315
Behavioral	0.472	2.120

**Table 6 ijerph-16-01203-t006:** Exploratory factor analysis of organizational cynicism and organizational pride.

Factor	Symbol	Loading	Variance Explained	EigenValue	Others Scales
Cognitive	Cog_1_	0.569	12.09	2.29	KMO = 0.892 Bartlett’s Test = 5331.044 Sig. = 0.000 Cumulative Variance = 70.69
	Cog_2_	0.637			
	Cog_3_	0.807			
	Cog_4_	0.579			
Affective	Aff_5_	0.682	18.80	3.57	
	Aff_6_	0.797			
	Aff_7_	0.805			
	Aff_8_	0.750			
Behavioral	Beh_9_	0.660	15.29	2.90	
	Beh_10_	0.654			
	Beh_11_	0.754			
	Beh_12_	0.705			
Emotional	Emo_13_	0.778	11.77	2.23	
	Emo_14_	0.820			
	Emo_15_	0.764			
	Emo_16_	0.559			
Attitudinal	Att_17_	0.899	12.74	2.42	
	Att_18_	0.844			
	Att_19_	0.858			

**Table 7 ijerph-16-01203-t007:** Model fit indices.

Indices	Symbol	Indices Values	Criteria
Chi-Square (*p* = 0.000)	(X^2^ = 399.443)	(DF = 143)	< 0.05
Chi-Square/Degree of Freedom	CMIN/DF	2.793	< 5
Root Mean Square of Approximation	RMSEA	0.072	< 0.08
Root Mean Square Residual	RMR	0.063	< 0.1
Comparative Fit Index	CFI	0.925	> 0.9
Tucker Lewis Inde	TLI	0.911	> 0.9
Incremental Fit Index	IFI	0.926	> 0.9
Normed Fit Index	NFI	0.909	> 0.9
Parsimony Normed Fit Index	PNFI	0.744	> 0.5
Goodness-of-Fit Index	GFI	0.905	> 0.9
Parsimony Goodness-of-Fit Index	PGFI	0.669	> 0.5

**Table 8 ijerph-16-01203-t008:** Hypothesis testing.

Hypo.	Structural Path	Estimate	S.E.	C.R.	P	Outcome
H_1_	Cog→Emo	0.157	0.127	1.238	0.216	Not Supported
H_2_	Aff→Emo	−0.353	0.122	−2.892	0.004	Supported
H_3_	Beh→Emo	−0.245	0.073	−3.364	***	Supported
H_4_	Cog→Att	0.028	0.201	0.138	0.890	Not Supported
H_5_	Aff→Att	−0.521	0.190	−2.737	0.006	Supported
H_6_	Beh→Att	0.142	0.112	1.268	0.205	Not Supported

Notes: *** = significance > 0.001, Critical ratio (CR > 1.96), then the path is significant at 0.05.

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
