# Peer review of "Organizational Cynicism and Its Impact on Organizational Pride in Industrial Organizations"

_ijerph, 2019, doi:10.3390/ijerph16071203_

Round 1

Reviewer 1 Report

This is an interesting and methodologically well put together paper which can make a useful contribution to the Special Edition: “Rethinking work in the digital era to protect the environment and promote health”.  However, there are a number of potential weaknesses that the authors might wish to consider in their revision. The grammar and use of English are generally poor throughout the paper and this needs to be dealt with through the aid of professional editing. The prevalence of organizational cynicism also needs to be indicated. What proportion of of the sample scored above the mean on the corporate cynicism scale? What were there biographical characteristics?  For example, were there more male than female above average scorers on organizational cynicism? Is organizational cynicism a widespread problem or a problem only suffered by a small proportion of the workforce.  Although the authors do try to link the focus of the study more with the theme of the special edition in Implications ( lines 347-362), a clearer linkage  could be made by briefly discussing the negative effects of cynicism on their mental health. Good luck in your revision. 

Author Response

Dear, 

Please, check attached file

Reviewer 2 Report

Review comments

[Title] Organizational Cynicism and its Impact on Organizational Pride in Industrial Organizations

1. The authors may want to move the objective of the research from the part of hypotheses to the introduction.

2. The authors may want to strengthen the causality of the cynicism and pride. There is no logical argument or theoretical support for the relationship between cynicism and pride.

3. The authors may want to include one or two sample items of the measurement in the revised manuscript or put an appendix which includes all measurement scales.

4. The authors may want to include some control variables into the analyses such as a gender.

5. The authors may want to defense (i.e., post hoc remedies) the common method bias (CMB) and include the possible problem of CMB in the limitation section.

6. The authors may want to conduct analysis the relationship between the total cynicism and the total pride in addition to the current analyses.

Author Response

Dear,

Please, check attached file

Reviewer 3 Report

The idea of the article is interesting especially in today’s context where there is a fierce competition on the market and for an organization each element can make a difference. It is well known that employees are the true treasure for an organization if they are connected to the value and objective of the organization. We are definitely heading to a change of mentality regarding work in general and employees in particular, so any research that brings knowledge in the field of relation between human resource and organizations are welcome.

The abstract of the paper has the main aspects needed in an article it brings the readers in the scope of the research and present the main findings. The innovativeness of the employees and how cynicism and trait influence it, is an interesting subject. So the premises for a good paper are fulfill. 

The introduction is good it allows readers to enter and familiarizes with the key concepts in this paper. The second part of the paper offers the necessary background for the research and is well divided. The hypotheses are well formulated and well presented. The third part of the paper is well written, I fund all the information that I need to understand how the study was conducted. For this part I only have a small recommendation, maybe a small paragraph can be introduced, about the survey data processing, and to explain what is and why do you chose AMOS 25.0 for this part. A short explanation of the tools used, with a short presentation of their advantages and if it is possible to indicate other tools that can be used in this cases. The fourth part of the paper, the results presentation is good. The fifth part is one of the most important part of the paper together with the abstract and conclusion, this is the most read part and it is very important to be well written.

As a general conclusion, I consider that the paper deals with an interesting topic of research and with small modification can bring a plus to the scientific field. The general aspect of the paper is a good one, the paper is well organized and thus it is easy to read, the references are from the last years. 

I hope that my recommendations are helpful for you and can better capitalize the work that you have done in this research. I wish you great success in future research.

Author Response

Dear,

Please, check attached file
